# arXiVeri: Automatic table verification with GPT

**Gyungin Shin**[1,4]        **Weidi Xie**[2,3]        **Samuel Albanie**[4]

[1]Visual Geometry Group, University of Oxford
[2]Cooperative Medianet Innovation Center, Shanghai Jiao Tong University
[3]Shanghai AI Laboratory
[4]CAML, University of Cambridge

## Abstract

Without accurate transcription of numerical data in scientific documents, a scientist cannot draw accurate conclusions. Unfortunately, the process of copying numerical data from one paper to another is prone to human error. In this paper, we propose to meet this challenge through the novel task of *automatic table verification* (*AutoTV*), in which the objective is to verify the accuracy of numerical data in tables by cross-referencing cited sources. To support this task, we propose a new benchmark, *arXiVeri*, which comprises tabular data drawn from open-access academic papers on arXiv. We introduce metrics to evaluate the performance of a table verifier in two key areas: (i) *table matching*, which aims to identify the source table in a cited document that corresponds to a target table, and (ii) *cell matching*, which aims to locate shared cells between a target and source table and identify their row and column indices accurately. By leveraging the flexible capabilities of modern large language models (LLMs), we propose simple baselines for table verification. Our findings highlight the complexity of this task, even for state-of-the-art LLMs like OpenAI's GPT-4. The code and benchmark is made publicly available.[1]

## 1 Introduction

Many areas of scientific research employ numerical data to analyse, summarise and communicate findings. When a researcher proposes a new framework, model or algorithm, it is often informative to compare their contribution with prior work by comparing performance metrics. These performance metrics are typically collated in tables that are interleaved with the body of text contained within scientific manuscripts. In practice, to enable the comparison, it is common for the researcher to manually copy performance metrics from the original manuscript into their own manuscript. While pragmatic, this copying process is susceptible to human error. When errors are introduced, the conclusions drawn from the comparisons are also affected. Given the importance of transferring such data correctly, there is a need for mechanisms that ensure its fidelity, but such tooling is not yet available. In short, we lack a "spell checker" for manually copied scientific data.

On first sight, the problem appears simple—after all, verifying that two numbers are equal is not a mathematically complicated task. However, in practice, it is beset with technical difficulties. Tables in the scientific literature are designed to be readable for a human audience rather than machine parsers. As such, they can vary significantly in layout, design, naming convention and manuscript location. The same numerical data may itself be reported at different levels of precision, using percentages, fractions or decimals and in absolute or relative metrics.

To meet these challenges we propose the task of *automatic table verification*—authenticating the numerical data encapsulated in tables by cross-verifying the referred sources. Specifically, we tackle this task with Large Language Models (LLMs) inspired by their strong performance in many text-based processing tasks [33, 31, 28, 18, 6, 19, 30, 21, 4, 22]. To facilitate evaluation of this task and address the incumbent challenges, we introduce *arXiVeri*, a succinct benchmark composed of tabular

---

[1]Code and benchmark are available at `https://github.com/caml-lab/research/tree/main/arxiveri`

NeurIPS 2023 AI for Science Workshop.

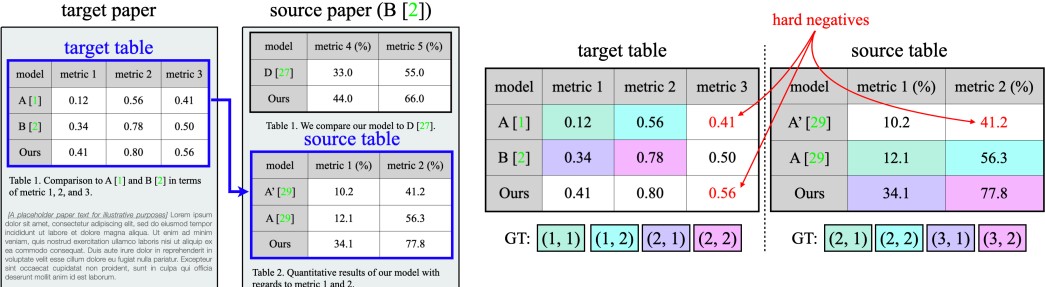

Figure 1: (Left) **Table matching**: given a target table from one paper and a list of source tables from another paper cited in the target table, the verifier needs to identify the source table containing numeric data, specifically floating point numbers, that supports the data presented in the target table. (Right) **Cell matching**: given a target table and a source table, the verifier needs to identify and locate cells that hold the same semantic content in both tables, subsequently outputting the respective row and column indices of these matching cells in each table. The cells that are emphasised in red depict the instances of hard negative cases. Best viewed in colour.

data extracted from open-access academic papers on arXiv. We further propose evaluation metrics for gauging the efficacy of the verification system in two key dimensions: *table matching* and *cell matching*. The former involves identifying the equivalent source table in a cited document for a given target table, while the latter aims to pair shared cells between a target and source table, and accurately identify their respective indices (see Fig. 1). Our experimental findings underscore the complexity inherent to the task (with frontier models such as GPT-4 [18] struggling in many cases), indicating that there is considerable room for further research progress.

Our contributions can be summarised as follows: (i) We introduce a new and challenging task called *Automatic Table Verification* (AutoTV), paving the way for advancements in automatic data verification in scientific documents; (ii) To stimulate further research in the AutoTV field, we introduce a benchmark dataset named *arXiVeri*, comprised of 3.8K target-source cell pairs and 158 target-source table pairs, sourced from publicly accessible papers on arXiv; (iii) To facilitate the assessment of AutoTV, we define a set of evaluation metrics for table matching and cell matching sub-tasks. In addition, we provide baselines to underpin future comparisons. (iv) Finally, we conduct a range of ablation studies to evaluate the key components of our approach which bring noticeable performance gains.

## 2 Related work

Our work is connected to large language models (LLMs) for scientific research, table detection and table structure recognition, and automating human labour with LLMs, which we describe next.

**Large language models for scientific research.** LLMs have been adapted for scientific research through various avenues, such as utilising models pretrained on scientific text to enhance performance in scientific NLP tasks [2, 29]. There are also notable advances in the biomedical sector, where specialised LLMs pretrained on biomedical text have demonstrated considerable improvements [25]. Additionally, the compilation of extensive academic paper corpora equipped with metadata and structured full text is proving to be an invaluable resource for academic research and text mining [16]. Alongside these advancements, there are investigations into the consequences of model scaling in scientific applications, evaluating the relationship between model size and performance [11]. Our research further develops this field by presenting a new challenge: automatic table verification in scientific documents, highlighting the essential role of data accuracy and integrity via cross-referencing cited sources.

**Tasks for tables in a single document.** Recent advancements in table-related tasks have primarily focused on detecting tables within documents and understanding their structure within a single document. Early efforts developed practical algorithms for detecting tables in heterogeneous documents [24], which later evolved with the incorporation of deep learning, specifically using Convolutional Neural Networks (CNNs), to enhance detection in PDF documents by combining visual features with non-visual information [10]. Subsequent research introduced end-to-end deep learning systems capable of not only detecting tables but also recognising their structure in document

images, without the need for metadata or heuristics [23]. Later work tackled both table detection and structure recognition simultaneously [20]. Prior work has also explored dataset construction for table extraction from unstructured documents [26]. However, the central theme of these works has remained the detection and structural understanding of tables within a single document. In contrast, we focus on table verification *across documents*.

**Task automation with LLMs.** LLMs have significantly impacted various automation tasks. For instance, Codex [5], an LLM fine-tuned on code from GitHub, exhibits proficient Python code generation capabilities, automating a task typically requiring human expertise. In the domain of data annotation, traditionally a labour-intensive task, LLMs like ChatGPT have demonstrated the potential to outperform human crowd-workers in speed, accuracy, and cost-effectiveness [8]. More recently, experiments have been conducted with GPT-4 to assess its ability to assist with Neural Architecture Search (NAS) [34] and interpreting neurons [3]. Our work also targets automation, offering a novel application of LLMs to automate the intricate task of table verification in scientific documents.

# 3 Automatic Table Verification

In this section, we define the proposed task of *automatic table verification* (Sec. 3.1) and metrics to evaluate the performance of a verifier on this task (Sec. 3.2). Then we describe our approach to tackle AutoTV (Sec. 3.3).

## 3.1 Task definition

The high-level objective of Automatic Table Verification (AutoTV) is to confirm that a document, referenced in a table (termed the target table) within a separate document, contains a corroborative table (termed the source table) which supports the cited information. When such a source table exists, AutoTV aims to identify matching cells between the source and target tables.

Our focus is particularly on instances within academic papers, where precise referencing of numeric data (e.g., floating point numbers) in tables is vital for comparative analysis. We observe that such in-table citations in academic literature occur for various reasons, including: attributing a specific approach to its original paper and quoting numerical data from an experimental result. The primary focus of table verification is the latter case, where a verifier is tasked with solving two sub-tasks (see Fig. 1): (i) **Table matching**: detecting a table in the cited document that matches a table in the referring document and if no such table exists, stating that there is no match; (ii) **Cell matching**: identifying correspondences between cells with a floating point number in the source and target tables that share the same semantic meaning. This implies not only identical numeric values, but also similar meanings as suggested by their respective table headers. The process includes pinpointing the location of such cells by providing their respective row and column indices in each table.

We note that these sub-tasks pose distinct challenges. First, there may not be a source table that matches the target table (e.g., the table citation may simply attribute to another document rather than quoting numbers). Second, multiple cells within a table (e.g., source table) can share the same numeric value, making it ambiguous how to pair those cells with ones in another table (e.g., target table). Third, a table can have a complex structure, with a single cell spanning across multiple rows and/or columns or featuring multiple headers, making it difficult to identify cell locations.

## 3.2 Evaluation metrics

To quantitatively measure performance of a verifier on AutoTV, we define four metrics including *table matching accuracy* for table matching, *cell matching recall*, *cell matching precision*, and *F-1 score* for cell matching as follows.

**Table matching accuracy** evaluates the verifier's ability to accurately identify a source table that matches a given target table, or to determine that no such source table exists in the cited document. Formally, given a set of all target tables $\mathcal{T}_t$, a target table $t \in \mathcal{T}_t$ with a set of $N_t$ in-table references $\mathcal{R}_t = \{r_i | 1 \le i \le N_t\}$, a set of candidate source tables $\mathcal{T}_{s;r_i}$ from a cited document $r_i$, and a verifier $\Phi(\cdot, prompt; \theta)$, the table detection accuracy (*Acc.*) is defined as:

$$Acc. = \frac{\sum\limits_{t \in \mathcal{T}_t} \sum\limits_{r_i \in \mathcal{R}_t} \delta[s_{r_i;t} = \hat{s}_{r_i;t}]}{|\mathcal{T}_t|}, \quad \hat{s}_{r_i;t} = \Phi(t, \mathcal{T}_{s;r_i}, prompt; \theta) \qquad (1)$$

where $\delta[\cdot]$, $\hat{s}_{r_i;t}$ and $s_{r_i;t}$ denote the Kronecker delta function, the detected source table and the ground-truth source table in the cited document which matches the given target table $t$, respectively.

| Target-source cell matching | |
| --- | --- |
| **Input** | a target table (`target_table`), a source table (`source_table`) |
| **System** | You are a helpful assistant. |
| **User** | Compare the following target and source tables and identify cells that contain floating point numbers with the same meaning present in both tables. Return the matched cells in a Python dictionary with the following format:

{
  (target_table_row_index, target_table_column_index):
  (source_table_row_index, source_table_column_index),
  ...
}

Use 0-based indexing, including headers, rowspan, and colspan attributes. Locate as many matching cell pairs as possible. If no matches are found, return an empty dictionary ({}).
The target table and its caption: `{target_table}`
The source table and its caption: `{source_table}` |
| **GPT-4** | `Answer` |

Table 1: **Text prompt used for the cell matching task.** We apply a regular expression to the answer string of the model to ensure the final result follows the specified Python dictionary format.

**Cell matching recall** quantifies the percentage of target-source cell matches that are accurately identified (i.e., true positives) among a ground-truth set of cell matches across a source table and a target table. Let us denote the ground-truth set of $N_{r_i;t}$ paired cells between a target table $t$ and a source table $s_{r_i;t}$ in a cited document $r_i$ as $\mathcal{C}_{r_i;t} = \{(c_t, c_{r_i;t})_j | 1 \leq j \leq N_{r_i;t}\}$ and a set of $\hat{N}_{r_i;t}$ detected cell matches as $\hat{\mathcal{C}}_{r_i;t} = \{(\hat{c}_t, \hat{c}_{r_i;t})_j | 1 \leq j \leq \hat{N}_{r_i;t}\}$ where $c_t$ and $c_{r_i;t}$ represent the row and column indices of a cell in the target and source tables, resp. Then, the cell matching recall ($Recall$) is defined as:

$$Recall = \frac{\sum\limits_{t \in \mathcal{T}_t} \sum\limits_{r_i \in \mathcal{R}_t} |\mathcal{C}_{r_i;t} \cap \hat{\mathcal{C}}_{r_i;t}|}{\sum\limits_{t \in \mathcal{T}_t} \sum\limits_{r_i \in \mathcal{R}_t} |\mathcal{C}_{r_i;t}|}, \ \hat{\mathcal{C}}_{r_i;t} = \Phi(t, s_{r_i;t}, prompt; \theta) \tag{2}$$

**Cell matching precision** measures how many target-source cell pairs are true positives among all the detected target-source cell pairs. Using the same notation as above, the cell matching precision ($Prec.$) is defined as:

$$Prec. = \frac{\sum\limits_{t \in \mathcal{T}_t} \sum\limits_{r_i \in \mathcal{R}_t} |\mathcal{C}_{r_i;t} \cap \hat{\mathcal{C}}_{r_i;t}|}{\sum\limits_{t \in \mathcal{T}_t} \sum\limits_{r_i \in \mathcal{R}_t} |\hat{\mathcal{C}}_{r_i;t}|}, \ \hat{\mathcal{C}}_{r_i;t} = \Phi(t, s_{r_i;t}, prompt; \theta) \tag{3}$$

**F$_1$ score** is a harmonic mean of the cell matching recall and precision to encapsulate both the measures in a single metric:

$$F_1 \ score = 2\frac{Prec. \times Recall}{Prec. + Recall} \tag{4}$$

**Remark.** All four metrics have a fixed range of [0, 1], with higher values being better. The text prompt, denoted by $prompt$, provided to the verifier may vary with the task, i.e., table matching and cell matching.

### 3.3 Baseline methods

To tackle AutoTV, we propose baseline approaches for table matching and cell matching as follows.

**Table matching.** We utilise a text embedding model (e.g., OpenAI's `text-embedding-ada-002`) to embed a target table alongside a set of candidate source tables from a document cited in the target table, including their respective captions. It is worth mentioning that the tables are in HTML format by default which we extract during the data collection process (detailed in Sec. 4.1). Subsequently, we rank the candidate tables based on their cosine similarities with the target table in the embedding space, selecting the one with the highest similarity score that also shares at least one

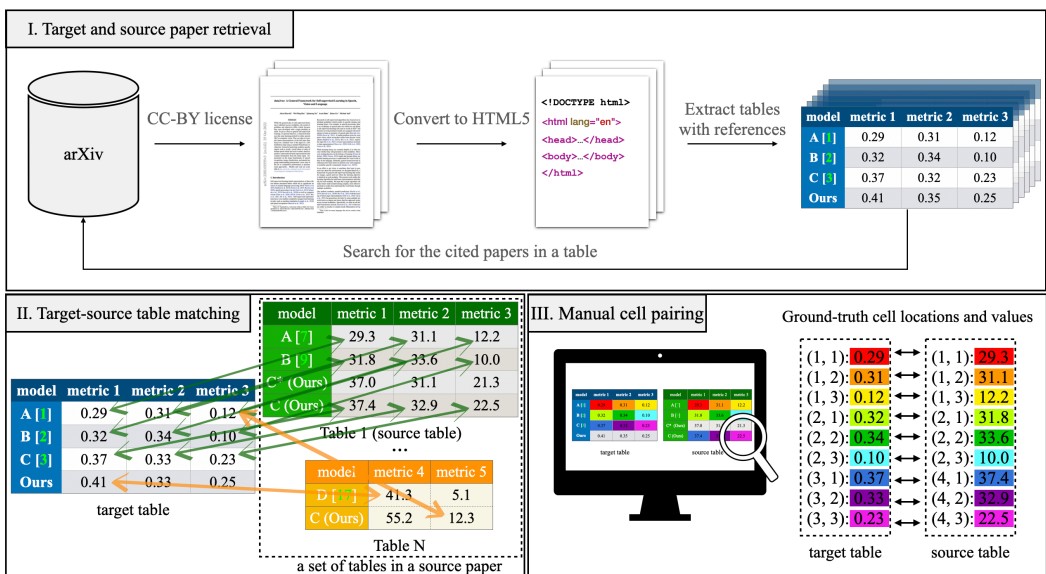

Figure 2: **Data collection pipeline for the arXiVeri benchmark.** Top: We randomly select open-access papers under the CC-BY license from arXiv and extract tables with in-table references (i.e. target tables) from an HTML5 version of the selected papers. Then, we repeat the process to retrieve the cited papers and their tables. Bottom left: To identify a *candidate* source table, which supports a target table, we pick one which has the most cells which are shared with the target table. Bottom right: Given the target and the candidate source table, we manually pair the common cells between them. If no paired cells are identified, we conclude that the candidate source table is a false positive and the source paper does not contain any matching source table for the target table. See the text for the details. Best viewed in colour.

floating point number with the target table. As the final step, we label the prediction as "no match found" if the chosen table's similarity score falls below a specified threshold.

In addition, we consider weighting each candidate table based on the number of floating point numbers that it shares with the target table before ranking them with their cosine similarity. In essence, we multiply the similarity score for a candidate table by a weight, which is determined by the number of floating-point numbers shared with the target table. Specifically, we sort the candidate tables based on the number of shared floats and assign each table with a weight between 0 and 1 according to their rank such that the table with the most shared floats is assigned with 1. These weights are evenly distributed with intervals of 1 divided by the count of candidate tables that share at least one floating point number with the target table. For tables that do not share any floating points, we assign a weight of 0. We show the effect of the weighting in Sec. 5.2.

Importantly, in all cases, floating point numbers are normalised to account for potential unit differences between the source and target tables.

**Cell matching.** We employ GPT-4 to extract matches between target-source cells containing a floating point number present in both target and source tables. As detailed in Sec. 3.2, each match is depicted as a pair of row and column indices for a cell in the target table and its corresponding cell in the source table. To facilitate this, we direct GPT-4 to generate a string representing a Python dictionary where keys denote cell indices in the target table and values represent indices in the source table as shown in Tab. 1. We then extract this dictionary using a regular expression that conforms to a specified dictionary pattern. For cell matching, we experiment with different types of commonly used table formats including HTML, CSV, or Markdown and show the effect of the table format in Sec. 5.2.

## 4 arXiVeri benchmark

Here, we introduce a benchmark composed of academic papers from arXiv, termed *arXiVeri*, for measuring performance of a verifier on the proposed AutoTV task. We first detail the data collection process (Sec. 4.1) and provide statistics of the arXiVeri benchmark (Sec. 4.2).

### 4.1 Data collection

As shown in Fig. 2, our data collection process is composed of three steps: (i) target and source paper retrieval, (ii) target-source table matching, and (iii) manual cell paring, as detailed next.

**Target and source paper retrieval.** We begin by collecting recent arXiv papers published in 2022 under the CC-BY license, using the open-source arXiv API.[2] We specifically focus on papers categorised as `cs.CV`, which had the highest number of submissions on arXiv in 2022. We extract tables along with their captions from each paper's HTML5 format using ar5iv,[3] which enables us to isolate tables from other elements such as main text by accessing the appropriate HTML tags (e.g., `<table>`). Subsequently, we employ an Elasticsearch-based arXiv search system to retrieve papers cited in each table (termed the source papers) with their title available in the "References" section of the referring paper (termed the target paper). As the title of a cited paper in the References section is often presented with irrelevant information (e.g., a url to a code) for the search system, we utilise `GPT-3.5-turbo` to extract the title from the whole reference information of a cited paper. Importantly, to increase the benchmark's complexity, we omit the cited paper if it does not contain a table sharing at least one cell value with the target table, or if it does not contain more than one table.

**Target-source table matching.** Given a table in a target paper (i.e., the target table) and a source paper which is cited in the target table, we select a table among the set of tables extracted from the source paper (using the same method described above) that supports the target table (i.e., the source table). Specifically, we choose a table that has the highest number of shared floating-point numbers with the target table to be the *candidate* source table. By iterating through all the references in a target table, we identify a corresponding candidate source table in each cited paper.

It is important to note that a target paper can have multiple tables referring to the same source paper, resulting in several potential table matchings between the target and source papers. In such cases, we choose the matching with the highest number of overlapping floating-point numbers per one target-source paper pair to increase the diversity of papers in the arXiVeri benchmark.

**Manual cell pairing.** In the final step of the collection process, we manually match cells that are commonly found in both target and candidate source tables. To determine a correct cell pair, we compare two cells from the tables and mark them as a match if they meet all of the following conditions:

(i) Both cells must represent the same value with an identical meaning, as indicated by their respective row and column headers.

(ii) Each cell must not contain more than one floating-point number for different metrics, avoiding the use of delimiters such as a comma (',') or a slash ('/').

(iii) If both cells have the same number of significant digits and the same unit, they must be exactly identical; for example, '12.3' and '12.4' would be treated as an incorrect pair.

The first condition ensures that matched cells have the same meaning as well as value, as determined by their row and column headers. The second condition aims to remove ambiguity during the evaluation step by avoiding cases where a single cell with multiple values is mapped to several cells in another table. The third condition accounts for potential discrepancies in rounding methods or mistakes, requiring matched cells to have the exact values given the same significant digits. If no such cell pairs are found between the target and candidate source tables, we regard the table pair does not have a source table from the source paper for the target table. On the other hand, if there is at least one cell pair, we treat the candidate source table as the source table (for the target table).

**Post-processing.** To ensure that the models used in our experiments can process each table and its caption as input, we filter out tables whose token length, including their captions, exceeds 3,072, as estimated by a tokeniser (i.e., `tiktoken`[4])

### 4.2 Statistics

We annotate a total of 3.8K cell pairs from 158 target-source table pairs, involving 110 different target papers and 158 distinct source papers. As illustrated in Fig. 3, we make three observations: (i) source papers contain an average of 4.6 tables, with three being the most frequent number of tables in a source paper; (ii) on average, there are 19.5 cell pairs between a target and a source table with the minimum and maximum number of cell pairs being 1 and 84, resp.; (iii) the dimensions of tables in the dataset exhibit a considerable range, with the smallest table measuring 4 by 5 and the largest reaching 20 by 19. On average, tables tend to fall around the size of 15.9 by 8.0.

---

[2] `https://github.com/lukasschwab/arxiv.py`

[3] `https://ar5iv.labs.arxiv.org`

[4] `https://github.com/openai/tiktoken`

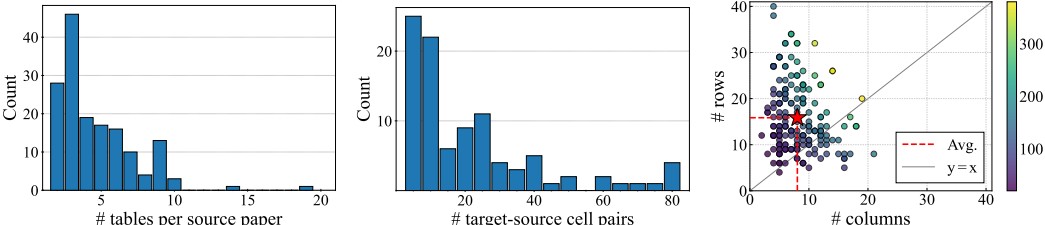

Figure 3: **Data statistics of the arXiVeri dataset.** Left: A histogram illustrating the count distribution of tables within source papers. Middle: A histogram representing 3.8K of shared cells between target and source tables. Right: A distribution plot of table dimensions (rows and columns), with colour indicating table size, and the average dimensions marked in red. Best viewed in colour.

## 5 Experiments

In this section, we first provide implementation details in Sec. 5.1 and conduct ablation studies in Sec. 5.2 to investigate each component of our approach for the proposed AutoTV task.

### 5.1 Implementation details

For the task of table matching, we employ four different text embedding models. These include OpenAI's `text-embedding-ada-002` which has an output dimension of 1536, as well as three models from Cohere: `embed-multilingual-v2.0`, `embed-english-v2.0`, and `embed-english-light-v2.0`, with respective output dimensions of 768, 4096, and 1024. For the cell matching task, we employ the `gpt-4-0314` model with a maximum length of 8,192 tokens and set the temperature parameter $\tau$ to 0, unless specified otherwise. To further minimize variability in the model's performance, we report the average score obtained by running each model three times across our experiments.

### 5.2 Ablation study

**Effect of text embedding models on table matching.** To investigate the influence of selecting different text embedding models, we evaluate four different models, as depicted in Tab. 2 (left). Alongside, we measure the performance of two baseline strategies: (i) "random", which selects a table from a candidate set of source tables, including "no match", and (ii) "overlap", which chooses a table that shares the most floating point numbers with the target table. As can be seen, each of the four embedding models significantly outperforms the baseline strategies by a margin of 12.7-15.8%. Among them, the `embed-english-light-v2.0` model from Cohere demonstrates the best performance.

**Effect of weighting on table matching.** As described in Sec. 3.3, we further refine our approach by weighting each candidate table based on the number of shared floating point numbers with the target table. Tab. 2 (right) illustrates the impact of this weighting mechanism on the performance of each embedding model. Notably, implementing this weighting strategy improves performance across all four embedding models, underscoring its effectiveness.

**Effect of table format and providing cell indices.** In the cell matching task, we explore three different table formats—HTML, CSV, and Markdown—for feeding tables to GPT-4. We posit that to enable the model to accurately identify a cell's location, providing explicit row and column indices for each cell could be beneficial. To verify this hypothesis, we also assess performance of the model when row and column indices are explicitly specified on the left and top of a table, resp.

From our results in Tab. 3 (left), we can see that the choice of format significantly influences the model's performance with the HTML format yielding the best performance in the absence of cell indices. We conjecture that this is because the HTML format contains more distinctive delimiters such as `<tr>` and `<td>` for table rows and table columns compared to CSV or Markdown where the model has to infer a cell location by counting a line break character and a comma (','), which can appear in other parts of the input than the actual table (e.g., text prompt and caption). Indeed, when cell indices are provided with an input table, we can observe that both of the CSV and Markdown formats have significant boost in all of the $Recall$, $Prec.$, and $F_1$ metrics, outperforming the HTML format. Examples of each format are provided in the supplementary materials.

**Effect of temperature.** In addition, we experiment with the temperature parameter in GPT-4, which modulates the randomness of the model's output. High values (nearing 1) introduce diversity, while low values (tending towards 0) enhance deterministic behavior. As shown in Tab. 3 (right), we

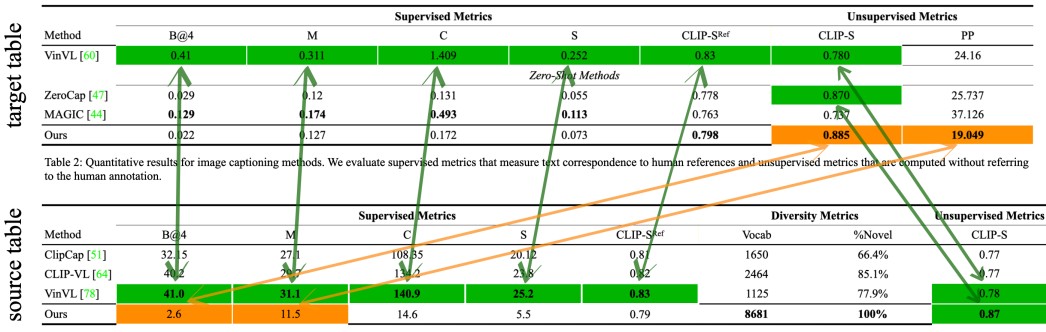

Table 2: Quantitative results for image captioning methods. We evaluate supervised metrics that measure text correspondence to human references and unsupervised metrics that are computed without referring to the human annotation.

| | | Supervised Metrics | | | | Unsupervised Metrics | |
|---|---|---|---|---|---|---|---|
| Method | B@4 | M | C | S | CLIP-S$^{Ref}$ | CLIP-S | PP |
| VinVL [60] | 0.41 | 0.311 | 1.409 | 0.252 | 0.83 | 0.780 | 24.16 |
| | | | *Zero-Shot Methods* | | | | |
| ZeroCap [47] | 0.029 | 0.12 | 0.131 | 0.055 | 0.778 | 0.870 | 25.737 |
| MAGIC [44] | **0.129** | **0.174** | **0.493** | **0.113** | 0.763 | 0.757 | 37.126 |
| Ours | 0.022 | 0.127 | 0.172 | 0.073 | **0.798** | **0.885** | **19.049** |

| | | Supervised Metrics | | | | Diversity Metrics | | Unsupervised Metrics |
|---|---|---|---|---|---|---|---|---|
| Method | B@4 | M | C | S | CLIP-S$^{Ref}$ | Vocab | %Novel | CLIP-S |
| ClipCap [51] | 32.15 | 27.1 | 108.35 | 20.12 | 0.81 | 1650 | 66.4% | 0.77 |
| CLIP-VL [64] | 40.2 | 29.7 | 134.2 | 23.8 | 0.82 | 2464 | 85.1% | 0.77 |
| VinVL [78] | 41.0 | 31.1 | 140.9 | 25.2 | 0.83 | 1125 | 77.9% | 0.78 |
| Ours | 2.6 | 11.5 | 14.6 | 5.5 | 0.79 | **8681** | **100%** | **0.87** |

Table 1. For each method, we report supervised metrics (i.e., ones requiring human references): B@1 = BLEU-1, M = METEOR, C = CIDEr, S = SPICE. We also report diversity metrics, which measures the vocabulary size (Vocab), and the number of novel sentences w.r.t the training set (%Novel). Finally, we report semantic relatedness to the image (CLIP-S), and to the human references (CLIP-S$^{Ref}$) based on CLIP's embeddings.

Figure 4: **A qualitative example of our approach for the cell matching task.** Cells marked in green denote accurate correspondences, while those highlighted in orange indicate mismatches.

| method | Inc. | dim. | *Acc.* |
|---|---|---|---|
| random | - | - | 13.7 |
| overlap | - | - | 27.2 |
| text-embedding-ada-002 | OpenAI | 1536 | 41.1 |
| embed-multilingual-v2.0 | Cohere | 768 | 39.9 |
| embed-english-v2.0 | Cohere | 4096 | 42.4 |
| embed-english-light-v2.0 | Cohere | 1024 | **43.0** |

| method | weighting | *Acc.* |
|---|---|---|
| text-embedding-ada-002 | ✗ | 36.1 |
| text-embedding-ada-002 | ✓ | 41.1 (+5.0) |
| embed-multilingual-v2.0 | ✗ | 34.2 |
| embed-multilingual-v2.0 | ✓ | 39.9 (+5.7) |
| embed-english-v2.0 | ✗ | 39.9 |
| embed-english-v2.0 | ✓ | 42.4 (+2.5) |
| embed-english-light-v2.0 | ✗ | 38.6 |
| embed-english-light-v2.0 | ✓ | 43.0 (+4.4) |

Table 2: **Ablation studies on the table matching task.** Left: we examine the impact of employing different text embedding models. We also offer random and overlap baselines denoted in grey (see the text for details). Right: we investigate the influence of implementing our proposed weighting function on the predictions generated by each embedding model.

| format | cell index | *Recall* | *Prec.* | $F_1$ score |
|---|---|---|---|---|
| HTML | - | 30.8 | 22.8 | 26.2 |
| CSV | ✗ | 21.2 | 16.2 | 18.4 |
| Markdown | ✗ | 18.1 | 13.5 | 15.4 |
| CSV | ✓ | 48.1 (+26.9) | 45.8 (+29.6) | 46.9 (+28.5) |
| Markdown | ✓ | 49.4 (+31.3) | 47.1 (+33.6) | 48.2 (+32.8) |

| $\tau$ | *Recall* | *Prec.* | $F_1$ score |
|---|---|---|---|
| 0.0 | 49.4 | 47.1 | 48.2 |
| 0.25 | **50.7** | 46.8 | 48.7 |
| 0.50 | 49.0 | 46.8 | 47.9 |
| 0.75 | 46.0 | 44.2 | 45.0 |
| 1.0 | 48.5 | **52.3** | **50.3** |

Table 3: **Ablation studies on the cell matching task.** Left: the impact of various input table formats, namely HTML, CSV, and Markdown and the influence of supplying cell indices for each table in the CSV and Markdown formats. Right: the effect of the temperature parameter for GPT-4 (using the Markdown format tables).

test temperatures of {0, 0.25, 0.5, 0.75, 1.0}, and find the optimal performance at a setting of 1.0, favouring diversity.

**Qualitative example.** In Fig. 4, we visualise GPT-4's predictions for the cell matching task. As can be noted, the model can map semantically identical cells despite unit differences in the target and source tables whereas it also predicts incorrect mappings between cells that represent different meanings (i.e., different metrics and methods). More examples are shown in the supplementary materials.

## 6 Conclusion

In this paper we address the critical task of ensuring numerical data accuracy in academic documents by introducing a novel task—*automatic table verification*—leveraging the capabilities of large language models. For this, we presented *arXiVeri*, a benchmark comprising tabular data from arXiv papers, and proposed metrics for evaluating verification performance. Despite the sophistication of advanced models like GPT-4, our findings underline the inherent complexity of the task, underscoring the necessity for further research in this field.

**Acknowledgements and disclosure of funding.** This work was performed using resources provided by the Cambridge Service for Data Driven Discovery (CSD3) operated by the University of Cambridge Research Computing Service (www.csd3.cam.ac.uk), provided by Dell EMC and Intel using Tier-2 funding from the Engineering and Physical Sciences Research Council (capital grant EP/T022159/1), and DiRAC funding from the Science and Technology Facilities Council (www.dirac.ac.uk). GS would like to thank Vishaal Udandarao for thorough proof-reading and Zheng Fang for the invaluable support. SA would like to acknowledge the support of Z. Novak and N. Novak in enabling his contribution. SA was supported by a Newton Grant.

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

# Appendices

In this supplementary material, we first describe the limitations (Sec. A) and broader impacts (Sec. B) of our work. Next, we provide additional statistics for the arXiVeri dataset in Sec. C and show examples of tables in different formats in Sec. D. Then we describe details of the prompts used for our ablation study in Sec. E and provide additional examples of predictions made by GPT-4 on the cell matching task in Sec. F. Lastly, we visualise an actual case where we find human errors in the process of quoting numeric data between tables in Sec. G.

## A   Limitations

We acknowledge several limitations to our approach. First, our task of automatic table verification currently only processes tables in text formats like HTML, CSV, or Markdown. This means our approach may not be suitable for table data embedded within images or PDF files, which are common formats in many documents. Second, the data collection pipeline for our arXiVeri benchmark is specifically designed to operate with arXiv papers that can be successfully transformed from PDF to HTML format via ar5iv. While this conversion allows us to cleanly extract tables with appropriate tags (e.g., `<table>`), this process may exclude certain papers if the conversion is unsuccessful, which could limit the diversity of table types included in the benchmark and potentially introduce a selection bias. Third, while the benchmark includes data from academic papers on arXiv, it may not fully encapsulate the variety of tables encountered across different domains. This could restrict the generality of our dataset. Future work should aim to extend our data collection pipeline to cater to a broader range of table sources. Lastly, we have limited insight into the GPT-4 inference process via OpenAI's API, and it is unclear if our encoded text is pre-processed or the model's output is post-processed. This can potentially affect the reproducibility of the experiments.

## B   Broader impacts

The automatic table verification proposed in this work has potential utility for many domains. In scientific research, it can reduce human error in data transcription and thus prevent such errors from influencing the interpretation of empirical data. In industries such as finance, healthcare, and engineering, it can ensure data accuracy, preventing costly mistakes.

Turning to negative impacts, the deployment of table verification (particularly while it remains far from perfect) may produce an elevated risk of over-reliance on the technology (with fewer "sanity checks" performed by researchers). More broadly, task automation may contribute to potential job displacement.

Finally, we note that care must be taken to mitigate privacy risks when deploying table verification across sensitive documents.

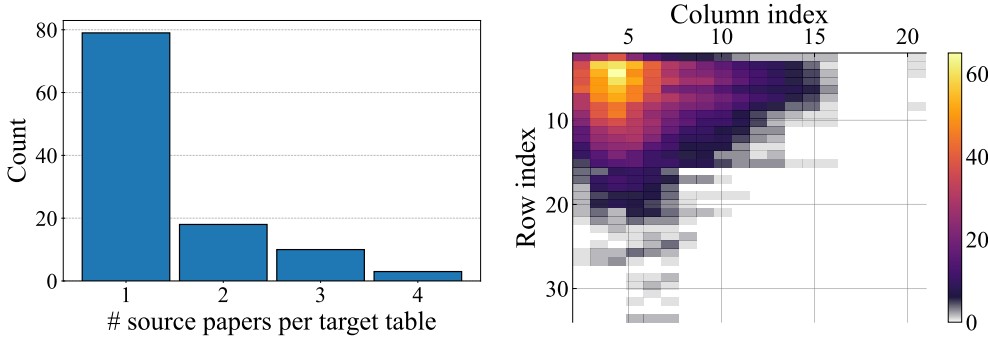

Figure 5: **Additional statistics for the arXiVeri bechmark.**

## C  Additional statistics

Here, we provide further details of statistics for the proposed benchmark, arXiVeri. In Fig. 5, we display the histogram of source paper counts per target table (left) and the distribution of matched cell locations in tables (right). We observe that (i) most target tables have a single source paper which is retrieved through the data collection pipeline (as decribed in Sec. 4 of the main paper) with the maximum number of source papers being 4; (ii) the locations of paired cells range from 1 to 33 for row indices, and from 1 to 20 for column indices, with an average cell location of approximately 7.3 and 4.9, respectively. Note that this is due to the fact that the average dimensions of the tables in the arXiVeri benchmark is 15.9 by 8.0, causing the average location of matching cells to fall short of the average dimensions of the tables.

**original table**

| Method | TGIF | | | MSRVTT | | LSMDC | | MSVD |
|---|---|---|---|---|---|---|---|---|
| | Action | Transition | Frame | MC | QA | MC | FiB | QA |
| **ClipBERT [18]** | 82.8 | 87.8 | 60.3 | 88.2 | 37.4 | - | - | - |
| **JustAsk [68]** | - | - | - | - | 41.5 | - | - | 46.3 |
| **MERLOT [19]** | 94.0 | 96.2 | 69.5 | 90.9 | 43.1 | 81.7 | 52.9 | - |
| **VIOLET** | 92.5 | 95.7 | 68.9 | 91.9 | 43.9 | 82.8 | 53.7 | 47.9 |

**HTML**

```
<table>
    <tr><td></td><td colspan=\"3\">TGIF</td><td colspan=\"2\">MSRVTT</td><td colspan=\"2\">LSMDC</td><td>MSVD</td></tr>
    <tr><td>Method</td><td>Action</td><td>Transition</td><td>Frame</td><td>MC</td><td>QA</td><td>MC</td><td>FiB</td><td>QA</td></tr>
    <tr><td>ClipBERT [18]</td><td>82.8</td><td>87.8</td><td>60.3</td><td>88.2</td><td>37.4</td><td>-</td><td>-</td><td>-</td></tr>
    <tr><td>JustAsk [68]</td><td>-</td><td>-</td><td>-</td><td>41.5</td><td>-</td><td>-</td><td>46.3</td></tr>
    <tr><td>MERLOT [19]</td><td>94.0</td><td>96.2</td><td>69.5</td><td>90.9</td><td>43.1</td><td>81.7</td><td>52.9</td><td>-</td></tr>
    <tr><td>VIOLET</td><td>92.5</td><td>95.7</td><td>68.9</td><td>91.9</td><td>43.9</td><td>82.8</td><td>53.7</td><td>47.9</td></tr>
</table>
```

**CSV**

```
,TGIF,TGIF,TGIF,MSRVTT,MSRVTT,LSMDC,LSMDC,MSVD
Method,Action,Transition,Frame,MC,QA,MC,FiB,QA
ClipBERT [18],82.8,87.8,60.3,88.2,37.4,-,-,-
JustAsk [68],-,-,-,-,41.5,-,-,46.3
MERLOT [19],94.0,96.2,69.5,90.9,43.1,81.7,52.9,-
VIOLET,92.5,95.7,68.9,91.9,43.9,82.8,53.7,47.9
```

**Markdown**

```
| nan           | TGIF    | TGIF       | TGIF  | MSRVTT | MSRVTT | LSMDC | LSMDC | MSVD |
| Method        | Action  | Transition | Frame | MC     | QA     | MC    | FiB   | QA   |
| ClipBERT [18] | 82.8    | 87.8       | 60.3  | 88.2   | 37.4   | -     | -     | -    |
| JustAsk [68]  | -       | -          | -     | -      | 41.5   | -     | -     | 46.3 |
| MERLOT [19]   | 94.0    | 96.2       | 69.5  | 90.9   | 43.1   | 81.7  | 52.9  | -    |
| VIOLET        | 92.5    | 95.7       | 68.9  | 91.9   | 43.9   | 82.8  | 53.7  | 47.9 |
```

Figure 6: **Examples of three different table formats used in our ablation study.** From top to bottom, the original table from [7], and the HTML, CSV, and Markdown formats are shown.

## D Examples of table formats

In Fig. 6, we display examples of three table formats considered in our ablation study: HTML, CSV, and Markdown. In the case of CSV and Markdown formats, table headers are reformatted to accommodate cells spanning multiple columns in the original table by repeating the cell value (i.e., TGIF, MSRVTT, and LSMDC).

| **Target-source cell matching** |
| --- |

| **Input** | a target table (`target_table`), a source table (`source_table`) |
| --- | --- |
| **System** | You are a helpful assistant. |
| **User** | Compare the following target and source tables and identify cells that contain floating point numbers with the same meaning present in both tables. Return the matched cells in a Python dictionary with the following format:

{
    (target_table_row_index, target_table_column_index):
    (source_table_row_index, source_table_column_index),
    ...
}

Use 0-based indexing. Locate as many matching cell pairs as possible. If no matches are found, return an empty dictionary ({}).
The target table and its caption: {`target_table`}
The source table and its caption: {`source_table`} |
| **GPT-4** | `Answer` |

Table 4: **Text prompt used for CSV and Markdown format tables on the cell matching task.**

| **Target-source cell matching** |
| --- |

| **Input** | a target table (`target_table`), a source table (`source_table`) |
| --- | --- |
| **System** | You are a helpful assistant. |
| **User** | Compare the following target and source tables and identify cells that contain floating point numbers with the same meaning present in both tables. Return the matched cells in a Python dictionary with the following format:

{
    (target_table_row_index, target_table_column_index):
    (source_table_row_index, source_table_column_index),
    ...
}

Use the row and column indices provided on the leftmost column and the topmost row of the tables, respectively. These indices are numerical and serve as identifiers to specify the location of each cell within the table. The row index is listed vertically along the left side of the table, while the column index is listed horizontally at the top. If no matches are found, return an empty dictionary ({}).
The target table and its caption: {`target_table`}
The source table and its caption: {`source_table`} |
| **GPT-4** | `Answer` |

Table 5: **Text prompt used for CSV and Markdown format tables *with cell indices* (on the cell matching task).**

# E  Prompts for cell matching

While the prompt shown in Sec. 3.3 of the main paper is used for tables in the HTML format, a slightly different prompt shown in Tab. 4 is utilised for the CSV and Markdown formats due to the absence of HTML attributes such as `colspan` (refer to Fig. 6 to see the differences between these table formats). For the case where we provide cell indices along with the tables, we employ the prompt as shown in Tab. 5.

| Method | R@1 | R@5 | R@10 | MdR |
|---|---|---|---|---|
| ClipBERT [21] | 22.0 | 46.8 | 59.9 | 6.0 |
| VLM [45] | 28.1 | 55.5 | 67.4 | 4.0 |
| MMT [9] | 26.6 | 57.1 | 69.6 | 4.0 |
| Support Set [34] | 30.1 | 58.5 | 69.3 | 3.0 |
| Frozen [2] | 31.0 | 59.5 | 70.5 | 3.0 |
| VideoCLIP [46] | 30.9 | 55.4 | 66.8 | - |
| HD-VILA [48] | 35.6 | 65.3 | 78.0 | 3.0 |
| Florence [52] | 37.6 | 63.8 | 72.6 | - |
| All-in-One [40] | 37.9 | 68.1 | 77.1 | - |
| BridgeFormer [11] | 37.6 | 64.8 | 75.1 | 3.0 |
| *CLIP-ViT-B/32* | | | | |
| CLIP4Clip [29] | 44.5 | 71.4 | 81.6 | 2.0 |
| CenterCLIP [57] | 44.2 | 71.6 | 82.1 | 2.0 |
| XPool [12] | 46.9 | 72.8 | 82.2 | 2.0 |
| CLIP2Video [7] | 45.6 | 72.6 | 81.7 | 2.0 |
| CLIP2Video† [3] | 47.2 | 73.0 | 83.0 | 2.0 |
| CLIP2TV [10] | 46.1 | 72.5 | 82.9 | 2.0 |
| DRL [43] | 47.4 | 74.6 | 83.8 | 2.0 |
| CAMoE* [6] | 47.3 | 74.2 | 84.5 | 2.0 |
| Ours | 50.1 | 74.8 | 84.6 | 1.0 |
| Ours* | 55.9 | 77.0 | 86.8 | 1.0 |

Table 5: Comparison of text-to-video retrieval in MSR-VTT [47]. * and † respectively denotes that the method uses DSL [6] and QB-Norm [3] as post-processing operations.

| Method | E2E† | Vis Enc. Init. | Visual-Text PT | # pairs PT | R@1 | R@5 | R@10 | MedR |
|---|---|---|---|---|---|---|---|---|
| JSFusion [75] | ✓ | - | - | - | 10.2 | 31.2 | 43.2 | 13.0 |
| HT MIL-NCE [44] | ✓ | - | HowTo100M | 136M | 14.9 | 40.2 | 52.8 | 9.0 |
| ActBERT [80] | ✓ | VisGenome | HowTo100M | 136M | 16.3 | 42.8 | 56.9 | 10.0 |
| HERO [34] | ✓ | ImageNet, Kinetics | HowTo100M | 136M | 16.8 | 43.4 | 57.7 | - |
| VidTranslate [28] | ✓ | IG65M | HowTo100M | 136M | 14.7 | - | 52.8 | |
| NoiseEst. [2] | ✗ | ImageNet, Kinetics | HowTo100M | 136M | 17.4 | 41.6 | 53.6 | 8.0 |
| CE [38] | ✗ | Numerous experts† | - | - | 20.9 | 48.8 | 62.4 | 6.0 |
| UniVL [40] | ✗ | - | HowTo100M | 136M | 21.2 | 49.6 | 63.1 | 6.0 |
| ClipBERT [32] | ✓ | - | COCO, VisGenome | 5.6M | 22.0 | 46.8 | 59.9 | 6.0 |
| AVLnet [55] | ✗ | ImageNet, Kinetics | HowTo100M | 136M | 27.1 | 55.6 | 66.6 | 4.0 |
| MMT [21] | ✗ | Numerous experts† | HowTo100M | 136M | 26.6 | 57.1 | 69.6 | 4.0 |
| T2VLAD [69] | ✗ | Numerous experts† | - | - | 29.5 | 59.0 | 70.1 | 4.0 |
| Support Set [48] | ✗ | IG65M, ImageNet | - | - | 27.4 | 56.3 | 67.7 | 3.0 |
| Support Set [48] | ✗ | IG65M, ImageNet | HowTo100M | 136M | 30.1 | 58.5 | 69.3 | 3.0 |
| Ours | ✓ | ImageNet | CC3M | 3M | 25.5 | 54.5 | 66.1 | 4.0 |
| Ours | ✓ | ImageNet | CC3M,WV-2M | 5.5M | 31.0 | 59.5 | 70.5 | 3.0 |
| Ours | ✓ | ImageNet | CC3M,WV-2M,COCO | 6.1M | 32.5 | 61.5 | 71.2 | 3.0 |
| Zero-shot | | | | | | | | |
| HT MIL-NCE | ✓ | - | HowTo100M | 136M | 7.5 | 21.2 | 29.6 | 38.0 |
| Support Set [48] | | IG65M, ImageNet | HowTo100M | 136M | 8.7 | 23.0 | 31.1 | 31.0 |
| Ours | ✓ | ImageNet | CC3M,WV-2M | 5.5M | 23.2 | 44.6 | 56.6 | 7.0 |
| Ours | ✓ | ImageNet | CC3M,WV-2M,COCO | 6.1M | 24.7 | 46.9 | 57.2 | 7.0 |

Table 4: Comparison to state-of-the-art results on MSR-VTT for text-to-video retrieval, 1k-A split. †E2E: Works trained on pixels directly, without using pre-extracted expert features trained for other tasks. Vis. Enc. Init.: Datasets used for pretraining visual encoders for tasks other than visual-text retrieval, eg object classification. Visual-Text PT: Visual-text pretraining data. Rows highlighted in blue use additional modalities such as sound and speech from the MSR-VTT test videos. †Object, Motion, Face, Scene, Speech, OCR and Sound classification features.

| Method | Model | PSNR ↑ | FM ↑ | $F_{ps}$ ↑ | DRD ↓ |
|---|---|---|---|---|---|
| Otsu [15] | Thres. | 9.74 | 51.45 | 53.05 | 59.07 |
| Savoula et al. [16] | Thres. | 13.78 | 67.81 | 74.08 | 17.69 |
| Kang et al. [2] | CNN | 19.39 | 89.71 | 91.62 | 2.51 |
| Tensmeyer et al. [25] | CNN | 19.11 | 88.34 | 90.24 | 4.92 |
| Zhao et al. [41] | cGA | 18.37 | 87.73 | 90.60 | 4.58 |
| Jemni et al. [3] | cGA | 20.18 | 92.41 | 94.35 | 2.60 |
| DocEnTr-Base{8} | Tr | 19.46 | 90.59 | 93.97 | 3.35 |
| DocEnTr-Base{16} | Tr | 19.33 | 89.97 | 93.5 | 3.68 |
| DocEnTr-Large{16} | Tr | 19.47 | 89.21 | 92.54 | 3.96 |

TABLE VII COMPARATIVE RESULTS OF OUR PROPOSED METHOD ON DIBCO 2018 DATASET. THRESH: THRESHOLDING, TR: TRANSFORMERS.

| Method | PSNR | FM | Fps | DRD | Avg |
|---|---|---|---|---|---|
| Otsu [10] | 9.74 | 51.45 | 53.05 | 59.07 | 38.79 |
| Sauvola et al. [11] | 13.78 | 67.81 | 74.08 | 17.69 | 59.50 |
| Adak et al. [2] | 14.62 | 73.45 | 75.94 | 26.24 | 59.44 |
| Souibgui et al. [5] | 16.16 | 77.59 | 85.74 | 7.93 | 67.89 |
| Tamrin et al. [6] | 17.04 | 83.08 | 88.46 | 5.09 | 70.87 |
| Zhao et al. [8] | 18.37 | 87.73 | 90.6 | 4.58 | 73.03 |
| Competition winner [2] | 19.11 | 88.34 | 90.24 | 4.92 | 73.19 |
| Akbari et al. [20] | 19.17 | 89.05 | 93.65 | 4.80 | 74.26 |
| Kang et al. [7] | 19.39 | 89.71 | 91.62 | 2.51 | 74.55 |
| Dang et al. [25] | 19.81 | 91.26 | 93.97 | 3.42 | 75.40 |
| Bera et al. [42] | 15.31 | 76.84 | 83.58 | 9.58 | 66.53 |
| Ours (S1) | 13.88 | 65.06 | 73.46 | 12.86 | 59.89 |
| Ours (S1) + Fine- | 20.18 | 92.41 | 94.35 | 2.60 | 76.08 |

Table 8: Results for all methods on H-DIBCO 2018 Dataset for handwritten document binarization. Avg = (PSNR + FM + Fps + (100 − DRD)) / 4.

Figure 7: **Two successful examples of cell matching predicted by GPT-4.** Even though there exist cells in both the target and source tables with identical values but different meanings as indicated by their respective table headers (marked in red), the model appropriately pairs only those cells that share both meaning and value. In both cases, the target tables are located on the left and the source tables on the right. The shaded area in the top left table denote cells that have been omitted for visual clarity. The tables are from [32, 1, 27, 13]. Best viewed in colour.

## F More qualitative examples

In Fig. 7, we present examples of successful applications of GPT-4 for cell matching, while Fig. 8 showcases two typical instances where GPT-4 failed to correctly perform the task. As observed in the successful cases, the model adeptly pairs cells from the target and source tables based on the shared meaning and value of the cells, even in the presence of hard negatives (i.e., cells highlighted in red) that exhibit the same value but differ in meaning.

In the failure cases, we observe that GPT-4 often matches cells based solely on their meanings, as defined by their respective table headers, despite the fact that the actual cell values between the pair differ. Another frequent type of error involves the model inaccurately locating cell indices.

| Method | COCO 1× | | COCO 2× | |
|---|---|---|---|---|
| | AP$^{bb}$ | AP$^{mk}$ | AP$^{bb}$ | AP$^{mk}$ |
| Supervised | 38.9 | 35.4 | 40.6 | 36.8 |
| MoCov2 [10] | 38.9 | 35.4 | 40.9 | 37.0 |
| BYOL† [23] | 40.6 | 37.5 | 42.0 | 38.7 |
| DenseCL [56] | 40.3 | 36.4 | 41.2 | 37.3 |
| ReSim [60] | 39.3 | 35.7 | 41.1 | 37.1 |
| PixPro [63] | 41.4 | - | - | - |
| DetCon$_B$† [28] | 41.5 | 38.0 | 42.1 | 38.9 |
| DetCo [61] | 39.4 | 34.4 | 41.4 | 35.8 |
| LEWEL [31] | 41.3 | 37.4 | 42.2 | 38.2 |
| R2O | 41.7 | 38.3 | 42.3 | 39.0 |

Table 1: Performance on COCO object detection and instance segmentation following ImageNet pretraining. All methods pretrained a ResNet-50 which later served as the backbone of a Mask R-CNN R50-FPN finetuned on train2017 for 12 epochs (1× schedule) or 24 epochs (2× schedule). We report Average Precision on bounding box (AP$^{bb}$) and mask (AP$^{mk}$) predictions for val2017. †: Results from re-implementation.

| Method | Fine-tune 1× | | Fine-tune 2× | |
|---|---|---|---|---|
| | AP$^{bb}$ | AP$^{mk}$ | AP$^{bb}$ | AP$^{mk}$ |
| Supervised | 39.6 | 35.6 | 41.6 | 37.6 |
| VADeR [48] | 39.2 | 35.6 | - | - |
| MoCo [24] | 39.4 | 35.6 | 41.7 | 37.5 |
| SimCLR [9] | 39.7 | 35.8 | 41.6 | 37.4 |
| MoCo v2 [11] | 40.1 | 36.3 | 41.7 | 37.6 |
| InfoMin [54] | 40.6 | 36.7 | 42.5 | 38.4 |
| PixPro [63] | 41.4 | - | - | - |
| BYOL [21] | 41.6 | 37.2 | 42.4 | 38.0 |
| SwAV [7] | 41.6 | 37.8 | - | - |
| DetCon$_S$ | 41.8 | 37.4 | 42.9 | 38.1 |
| DetCon$_B$ | 42.7 | 38.2 | 43.4 | 38.7 |

Table 2: Comparison to prior art: all methods are pretrained on ImageNet then fined-tuned on COCO for 12 epochs (1× schedule) or 24 epochs (2× schedule).

| Method | mIoU | OA | ceil. | floor | wall | beam | col. | wind. | door | chair | table | book. | sofa | board | clut. |
|---|---|---|---|---|---|---|---|---|---|---|---|---|---|---|---|
| PointNet [6] | 41.1 | 49.0 | 88.8 | 97.3 | 69.8 | 0.1 | 3.9 | 46.3 | 10.8 | 52.6 | 58.9 | 40.3 | 5.9 | 26.4 | 33.2 |
| KPConv (rigid) [10] | 65.4 | - | 92.6 | 97.3 | 81.4 | 0.0 | 16.5 | 54.5 | 69.5 | 90.1 | 80.2 | 74.6 | 66.4 | 63.7 | 58.1 |
| KPConv (deform) [10] | 67.1 | - | 92.8 | 97.3 | 82.4 | 0.0 | 23.9 | 58.0 | 69.0 | 91.0 | 81.5 | 75.3 | 75.4 | 66.7 | 58.9 |
| SPH-GCN [21] | 59.5 | 87.7 | 93.3 | 97.1 | 81.1 | 0.0 | 33.2 | 45.8 | 43.8 | 79.7 | 86.9 | 33.2 | 71.5 | 54.1 | 53.7 |
| SPNet [11] | 69.9 | 90.3 | 94.5 | 98.3 | 84.0 | 0.0 | 24.0 | 59.7 | 79.8 | 89.6 | 81.0 | 75.2 | 82.4 | 80.4 | 60.4 |
| Ours | 70.7 | 91.0 | 94.7 | 98.2 | 86.2 | 0.0 | 45.8 | 61.4 | 71.1 | 82.5 | 90.3 | 73.4 | 76.1 | 77.8 | 61.2 |

TABLE I: Semantic segmentation quantitative comparisons on S3DIS [73], tested on Area 5. We reported mean intersection over union (mIou) (%) and overall accuracy (%) scores as well as mIoU for individual classes.

| Method | mIoU | OA | ceil. | floor | wall | beam | col. | wind. | door | chair | table | book. | sofa | board | clut. |
|---|---|---|---|---|---|---|---|---|---|---|---|---|---|---|---|
| PointNet [23] | 41.1 | 49.0 | 88.8 | 97.3 | 69.8 | 0.1 | 3.9 | 46.3 | 10.8 | 52.6 | 58.9 | 40.3 | 5.9 | 26.4 | 33.2 |
| PointWeb [37] | 60.3 | 87.0 | 91.9 | 98.5 | 79.4 | 0.0 | 21.1 | 59.7 | 34.8 | 76.3 | 88.3 | 46.9 | 69.3 | 64.9 | 52.5 |
| Point2Node [7] | 62.9 | 88.8 | 93.8 | 98.3 | 83.3 | 0.0 | 35.6 | 55.3 | 58.8 | 79.5 | 84.7 | 44.1 | 71.1 | 58.7 | 55.2 |
| KPConv(R) [29] | 65.4 | - | 92.6 | 97.3 | 81.4 | 0.0 | 16.5 | 54.5 | 69.5 | 90.1 | 80.2 | 74.6 | 66.4 | 63.7 | 58.1 |
| KPConv(D) [29] | 67.1 | - | 92.8 | 97.3 | 82.4 | 0.0 | 23.9 | 58.0 | 69.0 | 91.0 | 81.5 | 75.3 | 75.4 | 66.7 | 58.9 |
| Ours | 69.9 | 90.3 | 94.5 | 98.3 | 84.0 | 0.0 | 24.0 | 59.7 | 79.8 | 89.6 | 81.0 | 75.2 | 82.4 | 80.4 | 60.4 |

Table 2: Semantic segmentation mIoU and OA scores on S3DIS [1] Area 5.

Figure 8: **Two common failure cases for cell matching are shown.** Green and orange arrows denote correct and incorrect cell matchings. Top: while GPT-4 paired the cells based on the meanings of the cells defined by their table headers, it failed to verify whether the cells also share the same value. Bottom: GPT-4 incorrectly positioned the cell indices in the target table (upper table) by shifting the matched cells one position to the right. The tables are from [9, 12, 14, 15].

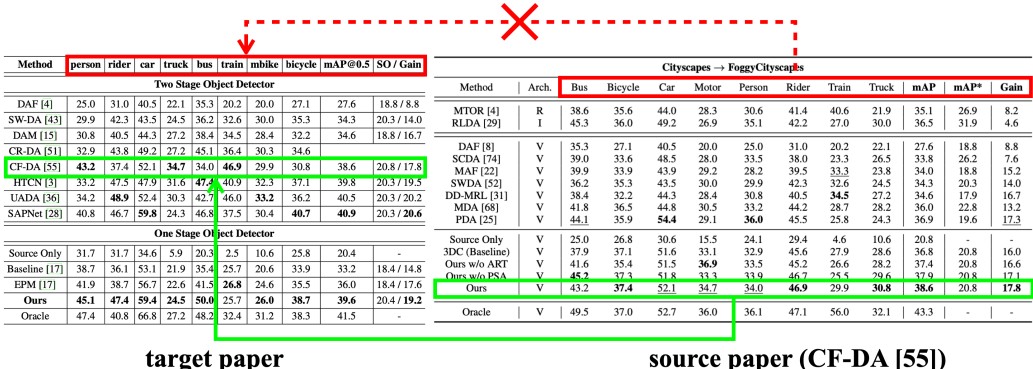

**target paper**         **source paper (CF-DA [55])**

Figure 9: **A practical example of a human error occurring during the process of transferring numerical data from one table to another.** Although the numbers from the source table [35] are accurately copied and pasted into the target table [17] while maintaining their original order (indicated in green), the sequence of the column headers in the target table has been altered (highlighted in red).

# G    Examples of human errors in transferring numeric data

Here, we visualise a practical example where we find human errors in the process of quoting numbers from a table to another. In Fig. 9, we note that the target table contains the numbers from the source table while keeping their order whereas the sequence of column headers is changed. We emphasise that our intention is not to cast blame on the authors of the target paper, but merely to highlight that such errors can inadvertently occur due to human oversight. This highlights the need for automatic table verification tools.

