# OpenReview forum: "arXiVeri: Automatic table verification with GPT"
_NeurIPS.cc/2023/Workshop/AI4Science — NeurIPS2023-AI4Science Poster_

### Official Review · Reviewer_kVpx · 2023-10-25
**arXiVeri: Automatic table verification with GPT**

**Rating:** 6
**Confidence:** 3

**Review:**

There are lot of assumptions in the method. The authors need to edit the abstract and conclusions highlighting where their method will fail. The limitations are listed at the end, which makes the current version of the manuscript over-claiming the actual results. I wonder if the method can work with other papers from Bioarxiv or Medarxiv?

---

### Meta-Review · Area_Chair_wXhf · 2023-10-27

**Recommendation:** Accept (Poster)
**Confidence:** 4

**Metareview:**

The paper newly introduce a task called automatic table verification (AutoTV), and introduces arXiVeri benchmark. The authors propose several metrics, i.e., table matching accuracy, cell matching recall, cell matching precision, and F1, to evaluate the table verification performance. Despite several drawbacks raised by the reviewers, the proposed work has its own merit in that it tried to define a new task that is important in this domain. I suggest the authors to evaluate the proposed method on other papers from Bioarxiv or Medarxiv.